# Supplementation for Performance and Health in Patients with Phenylketonuria: An Exercise-Based Approach to Improving Dietary Adherence

**DOI:** 10.3390/nu16050639

**Published:** 2024-02-25

**Authors:** Domingo González-Lamuño, Carmenmelina Morencos, Francisco J. Arrieta, Eva Venegas, Germán Vicente-Rodríguez, José Antonio Casajús, Maria Luz Couce, Luís Aldámiz-Echevarría

**Affiliations:** 1University Hospital “Marqués de Valdecilla”, Universidad de Cantabria and Research Institute Valdecilla (IDIVAL), 39008 Santander, Spain; 2Exercise and Health in Special Population Spanish Research Net (EXERNET), 50009 Zaragoza, Spain; gervicen@unizar.es (G.V.-R.); joseant@unizar.es (J.A.C.); 3Asociación Española para el Estudio de los Errores Congénitos del Metabolismo (AECOM)—AECOM&Sociedad, 28221 Majadahonda, Spain; arri68@hotmail.com (F.J.A.); evamvenegas@gmail.com (E.V.); maria.luz.couce.pico@sergas.es (M.L.C.); jixalazl@hotmail.com (L.A.-E.); 4Danone Nutricia Metabolics, 28043 Madrid, Spain; carmen-melina.morencos@danone.com; 5Instituto Ramón y Cajal de Investigación Sanitaria (IRYCIS), 28034 Madrid, Spain; 6Unidad de Gestión Clínica de Endocrinología y Nutrición, Instituto de Biomedicina de Sevilla (IBiS), Hospital Universitario Virgen del Rocío, CSIC, Universidad de Sevilla, 41013 Seville, Spain; 7Instituto de Investigación Sanitaria Aragón (IIS Aragón), 50009 Zaragoza, Spain; 8EXER-GENUD (Growth, Exercise, Nutrition and Development) Research Group, Universidad de Zaragoza, 50009 Zaragoza, Spain; 9Centro de Investigación Biomédica en Red de Fisiopatología de la Obesidad y Nutrición (CIBERObn), 28040 Madrid, Spain; 10Faculty of Health and Sport Sciences, FCSD, Ronda Misericordia 5, 22001 Huesca, Spain; 11Unit for Diagnosis and Treatment of Congenital Metabolic Disorders, University Hospital of Santiago de Compostela, 15706 Santiago de Compostela, Spain; 12Instituto de Investigación Sanitaria de Santiago de Compostela, 15706 Santiago de Compostela, Spain

**Keywords:** phenylketonuria, protein substitutes, amino acid, glycomacropetides, phenyalanine, tyrosine, physical activity, supplement regimens, adherence

## Abstract

Supplementation is crucial for improving performance and health in phenylketonuria (PKU) patients, who face dietary challenges. Proteins are vital for athletes, supporting muscle growth, minimizing catabolism, and aiding muscle repair and glycogen replenishment post-exercise. However, PKU individuals must limit phenylalanine (Phe) intake, requiring supplementation with Phe-free amino acids or glycomacropeptides. Tailored to meet nutritional needs, these substitutes lack Phe but fulfill protein requirements. Due to limited supplement availability, athletes with PKU may need higher protein intake. Various factors affect tolerated Phe levels, including supplement quantity and age. Adhering to supplement regimens optimizes performance and addresses PKU challenges. Strategically-timed protein substitutes can safely enhance muscle synthesis and sports performance. Individualized intake is essential for optimal outcomes, recognizing proteins’ multifaceted role. Here, we explore protein substitute supplementation in PKU patients within the context of physical activity, considering limited evidence.

## 1. Introduction

Phenylketonuria (PKU) is an inherited disorder that affects phenylalanine (Phe) metabolism. It is caused by Phe hydroxylase enzyme (PAH) deficiency. Treatment with a low-Phe diet, often supplemented with protein substitutes, is necessary to prevent Phe accumulation in the brain [1,2]. Since patients with PKU cannot convert Phe into tyrosine, they need to restrict their daily protein intake and take commercial preparations of Phe-free amino acid or glycomacropeptide supplements tailored to variables such as age and patient status [3].

The 1994 Dietary Supplements Health and Education Act (DSHEA) definition of a dietary supplement is inadequate for PKU, as it focuses on a “healthy diet.” Patients with PKU, however, have dietary needs distinct from those of the general population, and are dependent on supplementation with protein substitutes. These products are ingested purposefully and are designed to complement a regular diet by supplying the nutrients necessary to achieve specific health and performance benefits. They come in various forms, such as functional foods, formulated foods, sports foods, single nutrients, and multi-ingredient products [4].

A “healthy diet” for a patient with PKU varies according to the amount of Phe they can tolerate. Dietary Phe tolerance is influenced by numerous factors, including PKU severity, age, growth rate, pregnancy, and catabolic states associated with intercurrent illness. It is also influenced by the type and protein substitute dosage, pharmacological treatments (BH4), and type and intensity of physical activity.

Overall, individuals with PAH have similar nutrient requirements to the general population. However, they may have higher protein needs due to reduced protein availability in protein substitutes and possible mitigation of protein catabolism by supplements.

A range of commercially available low-protein food products, such as bread, pasta, flour, and cookies can enhance dietary adherence in PKU [3]. Dietary interventions play a crucial role in the management of PKU and help ensure overall well-being. Although adolescents with severe PKU are potentially prone to excessive weight gain [5], optimal physical development is an achievable goal in PKU, regardless of phenotype or severity of dietary restrictions [6].

In athletes with PKU, optimal nutritional management is challenging but feasible. An exercise-based approach tailored to individual needs can help improve adherence to dietary recommendations and protein substitutes, which are crucial for both performance and health. Here, we review the use of protein substitute supplementation in PKU patients in the context of physical activity. 

In selecting articles for inclusion, the criteria primarily focus on relevance to the topic, quality of research, currency, diversity of perspectives, applicability, and consensus among authors. Articles must directly address PKU management, exercise, and sports nutrition, preferably from reputable peer-reviewed journals and recent publications. The search strategy involves using relevant keywords in scientific databases and considering citation tracking and reference lists. Consensus among authors is reached through open communication, mutual agreement, and evaluation of each article’s relevance and contribution to the paper’s objectives.

## 2. Exercise and PKU

Exercise is a crucial component of a healthy lifestyle, and individuals with PKU are encouraged to engage in regular physical activity in line with World Health Organization (WHO) guidelines [6]. Recognizing the distinction between physical activity, exercise, and sport is essential for understanding the spectrum of bodily movements that people engage in daily and the impact these have on overall well-being. Physical activity is defined as any bodily movement produced by skeletal muscles that results in energy expenditure. Exercise is a subset of physical activity that is planned, structured, and repetitive and has as a final or intermediate goal the improvement or maintenance of physical fitness and health [7,8]. Sport, in turn, is a rule-based subset of physical activity involving physical exertion and skill that is undertaken by individuals or teams for recreational or competitive purposes. 

Although patients with well-treated PKU should be able to engage in physical activity, including exercise and sport, this endeavor comes with increased nutritional requirements and necessitates dietary adjustments. The primary goal is to maintain optimal metabolic control of Phe levels while achieving maximal athletic performance. The benefits of physical exercise extend beyond the muscular system, impacting cardiovascular capacity, bone health, and cognitive processes, and overall contributing to improved self-perception and self-esteem.

Specific sports nutrition come into play for individuals with PKU. Key objectives are to maintain a high carbohydrate diet, closely monitor hydration status, and strategically time protein substitute intake in the immediate post-exercise recovery phase. Optimal energy intake, with careful selection of low-protein foods, is essential before, during, and after exercise or competition. While acute exercise does not appear to influence blood Phe concentrations, the impact of endurance exercise has not been examined [9]. Nevertheless, exercise has been linked to overall health benefits in PKU patients [9,10] (Table 1).

Rocha et al. [10] outlined a series of critical considerations for individuals with PKU engaging in physical exercise. Depending on the intensity and duration of the activity, practitioners may need additional energy intake or a combination of energy and amino acid supplements to mitigate catabolism-induced decompensation. Dietary adjustments should ideally be guided by the specific goals of the exercise regimen, which could be fat reduction or muscle gain. Phe-free protein substitutes are recommended for muscle mass enhancement, with guidance from a dietitian or attending physician to clarify uncertainties regarding dietary adaptations. A reduction in energy supplementation is advised when the goal is weight loss, but complete cessation is not recommended as it could cause decompensation. Hydration is critical, with water recommended as the primary means of ensuring optimal fluid balance during physical activity. The above considerations highlight the need for individualized dietary strategies and professional guidance for individuals with PKU engaging in exercise or sport [10].

## 3. Protein Metabolism and Exercise 

Protein metabolism during exercise is a multifaceted process, with proteins serving various physiological functions: they form the structural basis of muscle tissue, are essential components of muscle enzymes, contribute to the immune system, and play a crucial role in overall physical performance [17]. Several parameters and techniques are employed to study protein metabolism during exercise. Urea concentrations and 3-methylhistidine levels in urine serve as indicators of protein metabolism, while analysis of nitrogen balance offer a comprehensive assessment of protein status [18]. Experimental tracing studies with radioactive isotope-labeled amino acids provide detailed insights into metabolic functions, although their utility is limited by their invasive nature, potential hazards associated with isotope use, and high costs [17,18].

While proteins are not considered a primary source of energy during physical activity, they are important for endurance sports or long workouts, where there is a risk of muscle and hepatic glycogen store and intramuscular fat depletion. In such scenarios, proteins can serve as an important source of energy, contributing to up to 5% of total energy expenditure [19,20]. Amino acids, in particular branched-chain amino acids (BCAAs), are also valuable during prolonged exercise, as they can supply up to 10–15% of energy required [21]. Protein catabolism intensifies following substantial depletion of muscle glycogen stores (33–55%), especially in activities that cause microtraumas. This catabolic process generates amino acids, such as leucine, which can enter energy pathways, producing ketone bodies. Other amino acids, such as valine and isoleucine, contribute to gluconeogenesis and become essential energy sources when glycogen stores are diminished during prolonged exercise [22].

Carbohydrates and fats have traditionally been recognized as the primary sources of energy during physical activity. Proteins also contribute to energy production, but their primary role is to support muscle structure, function, and repair [17]. Ensuring adequate intake of these macronutrients is crucial to fulfill energy requirements and optimize overall athletic and exercise performance.

## 4. Daily Use of PKU Protein Substitutes

Provision of an adequate dose of protein substitute, typically comprising essential amino acid supplements devoid of Phe and enriched in tyrosine or glycomacropeptides, is crucial to promote normal growth, prevent protein deficiency, provide a source of tyrosine, and optimize control of blood Phe levels. For individuals with classic PKU, protein substitutes are likely to supply at least 70–75% and up to 80% of daily nitrogen requirements [1].

European guidelines on the diagnosis and treatment of PKU recommend a total protein intake of 40% higher than that recommended by the Food and Agricultural Organization and the WHO, although this difference is not supported by research. Increased protein intake through protein substitutes compensates for inefficient absorption of natural/intact protein (mainly of plant origin), poor utilization of L-amino acids, and suboptimal energy consumption [2].

Supplement intake results in a rise in plasma amino acid concentrations and a faster decline, leading to increased loss. It is therefore recommended to administer protein substitutes in small but frequent doses (3–4 doses), spread evenly throughout the day, along with natural protein and a source of carbohydrates.

### Types of Protein Substitutes

Amino acid-based protein substitutes. Amino acid-based protein substitutes are available as powders, capsules, tablets, bars, and liquids, and may also contain carbohydrates, fats, vitamins, and minerals. Adherence can be challenging, and it is therefore important to offer nutritionally appropriate presentations that are suitable for the individual’s profile and age. 

Amino acid-based protein substitutes have high osmolality (expressed as the total number of solute particles per kilogram). Products with high osmolality can cause delayed gastric emptying and diarrhea. Additional water is thus recommended with each dose of protein substitute. Tyrosine is hydrophobic and forms an insoluble top layer. Solutions should therefore be shaken well to achieve a homogeneous mixture. Ideally, all protein substitutes should be prepared immediately before use. Some preparations contain starch that thickens over time. Products should be stored in a cool place and expiry dates checked.

Glycomacropeptides. Glycomacropeptide (GMP) is a low-Phe protein used as a protein substitute for PKU. It is a byproduct of cheese whey. Although theoretically Phe-free, some residual Phe, remains in the manufactured product due to the extraction process. Glycomacropeptides (±vitamins and minerals) are available in powder, liquid, and bar forms and provide 1.8 mg of Phe per gram of protein equivalent. According to European PKU guidelines, additional testing is required before practical recommendations can be made on the use of glycomacropeptides in PKU [2].

## 5. Diet and Protein Substitutes: Insights from Sports Medicine

### 5.1. Dietary Supplements

A dietary supplement, as defined by Maughan et al. [23], serves as nutritional support, providing energy and nutrients distinct to those found in regular foods. Dietary supplements are widely used by athletes across different levels of sports. Approximately 50% of the adult population in the United States have reported using some form of dietary supplementation, and prevalence is likely similar in many other countries [23]. The use of nutritional supplements is particularly high among elite athletes [24].

Dietary supplements are regulated products, but in many countries regulations are often lax or loosely enforced, leading to the distribution and marketing of products with unverified attributes that may fall short of labeling or composition standards. In contrast to pharmaceutical products, dietary supplements are not typically subject to rigorous controls [25]. In Australia, which has comprehensive regulations in this domain, supplements are categorized into four groups [26]:Group A: Approved supplements with scientifically proven energy-boosting properties, nutrients, and benefits.Group B: Supplements of potential interest that require further studies due to insufficient evidence.Group C: Potentially harmful supplements with no evidence of improved sports performance.Group D: Banned supplements associated with doping practices.

The above classification system underscores the need for discernment and caution in the use of dietary supplements and the importance of scientific validation and adherence to regulatory standards to ensure the efficacy and safety of sports foods and nutritional supplements.

### 5.2. Protein Intake for Athletes

Protein requirements for athletes are influenced by age, health status, and type of activity. In sedentary individuals, the percent range of total calories from proteins to maintain nitrogen balance is 8–10%, but in athletes this proportion can almost double (15–20%) [27]. The recommended protein intake for muscle mass ranges from 1.2–2.2 g per kilogram of body weight per day [28]. These figures highlight the importance of assessing whether nutritional needs can be met through diet alone. Moreover, the question arises whether supplementation is necessary for endurance and strength athletes or if personalized diets can be sufficient to ensure appropriate amino acid intake.

It has been suggested that athletes undergoing intense training may require 1.7–2.2 g/kg of protein per day, and whey protein shows benefits when consumed in doses of 20–40 g every 3–4 h; whey protein promotes muscle protein synthesis and enhances body composition and performance [29]. The timing of protein intake is crucial, with benefits observed especially after exercise [30].

Protein supplements such as whey protein, casein-derived proteins, bovine colostrum-derived proteins, soy-derived proteins, and egg proteins are all considered high-quality proteins, and have distinct characteristics. Casein, for example, is a slow-release protein, whereas whey protein and protein hydrolysates are absorbed rapidly [31].

The safety of protein supplementation is established within tolerable limits, but evidence on potential harm to the kidneys or liver is inconclusive. Excessive protein intake can have adverse consequences, but recommended doses (up to 2.2 g/kg per day) pose no risk to healthy athletes.

Protein supplementation positively influences body composition and sports performance, particularly when combined with strength or resistance training. While protein intake exceeding 3 g/kg per day may not enhance sport performance, it can positively affect the body composition of endurance athletes [28]. When combined with a diet with adequate energy and nutrient intake, protein supplementation appears to increase muscle strength, especially when taken at high doses (>2 g/kg per day), slightly enhance performance in endurance sports, stimulate muscle protein synthesis, and aid recovery.

### 5.3. Protein Supplements

Nutritional supplements are widely used among athletes. The most popular products on the market are protein powders, followed by BCAAs in capsule and powder form [26]. BCAAs have gained popularity due to their purported benefits, including fatigue response improvement and immune system support for endurance athletes and enhanced essential amino acid availability for strength athletes [32,33].

BCAAs, which comprise leucine, isoleucine, and valine, play an essential role in protein synthesis and energy regulation. Despite their popularity for reducing muscle damage and enhancing various physiological parameters, dosage recommendations remain unclear [34]. For optimal athletic performance, branched-chain amino acids (BCAAs) are commonly recommended in dosages ranging from 5 to 10 g before, during, and after exercise. This dosage regimen aims to support muscle protein synthesis, reduce muscle breakdown, delay fatigue, and promote muscle recovery and repair. Studies suggest that BCAA supplementation may enhance exercise performance, reduce muscle soreness, and improve muscle adaptation to training [35,36]. However, individual responses to supplementation can vary, and it is important to start with lower doses and gradually increase as needed while monitoring for adverse effects [37]. A 2:1:1 ratio is recommended for these proteins, but their impact on competitive performance remains to be determined [38]. Leucine, known for its anabolic effects, aids muscle recovery and is well tolerated [39].

Various amino acids and nitrogenous substances can contribute to athletic performance [40]. β-alanine improves muscle contraction and buffering capacity, which is particularly beneficial for exercise lasting 1–4 min. Taurine, with insulin-mimicking effects, supports immune function and glycogen loading and is well tolerated at recommended doses. Glutamine prevents protein breakdown and fatigue and is recommended for athletes at a safe dose of 5–10 g per day. Arginine, a nitric oxide precursor, is supported by limited evidence and should be used with caution. Evidence on the benefits of aspartic acid in endurance exercise is also inconclusive, but the recommended daily limit is 3 g. The daily limit for choline is 1.5 mg and should ideally be met through diet, although the ergogenic effects remain unclear. Glycine, which is associated with joint health, requires further study, and again caution is advised. Daily intake of inosine should not exceed 450 mg and inconsistent results call for careful use [40].

Creatine is a well-demonstrated ergonutritional supplement, notably effective in explosive power sports, high-intensity short-duration activities, and muscle hypertrophy training [41,42]. Caution, however, is warranted in aerobic endurance sports due to possible fluid retention.

Although studies suggest that athletes have higher protein requirements, natural food sources are recommended over supplements [43,44,45,46]. High-performance athletes, however, may find it difficult to achieve adequate intake without supplementation. Protein hydrolysates may aid absorption and muscle regeneration but are most effective in cases of dietary deficiencies and positive energy intake [45,47,48].

### 5.4. Calculation of Protein Kinetics for Patients with PKU

Individuals with PKU are heavily reliant on routine protein substitutes in the form of amino acids or glycomacropeptides. These account for over 70% of their protein requirements and are essential sources of energy, vitamins, and minerals [2]. Vitamin B12 deficiency is relatively common in patients with PKU compared to other micronutrient deficiencies. This is because the main source of vitamin B12 is protein-rich food, which is restricted in the PKU diet. Vitamin B12 deficiency is mainly reported in adolescent and adult PKU patients who do not regularly take Phe-free amino acid supplements including vitamins [49].

To assess the impact of protein substitutes on sports performance, it is first necessary to understand whole-body protein dynamics. The intricate calculation of protein kinetics plays a pivotal role in evaluating protein synthesis, breakdown, and net balance, especially in the post-absorptive to fed-state transition [50,51]. In the post-absorptive state, protein breakdown is computed based on the rate of Phe appearance, a process intricately linked to the determination of protein synthesis. In the transition to the post-prandial state, it is necessary to factor in the contribution of dietary Phe to the overall appearance of Phe in the bloodstream. This nuanced consideration is essential for accurate estimation of protein synthesis and net protein balance during the post-meal period [52]. 

This methodological overview sheds light on the intricacies of calculating whole-body protein kinetics and provides a comprehensive perspective on synthesis, breakdown, and net balance processes. Deficiency in PAH in individuals with PKU further complicates the interpretation of these processes [53]. 

## 6. Supplements for Performance and Health in Patients with PKU

The primary consideration for physically active individuals with PKU is to ensure that protein requirements are met. Additional supplementation (with BCAAs, creatine, caffeine, etc.) should only be considered under specific circumstances, as the specially designed PKU preparations or substitutes providing Phe-free protein equivalents are the principal source of protein for patients with PKU. Given the restriction of protein-containing foods, including traditional sources of creatine, creatine supplementation could become particularly relevant for individuals with PKU who are engaged in athletic pursuits.

Protein substitutes designed for PKU fall under the category of food for special medical purposes and should be used under medical or nutritional supervision. Depending on the severity of the patient’s condition, PKU preparations can provide up to 80% of daily protein requirements. The remaining proteins are obtained from regular meals, customized to accommodate individual Phe tolerance levels. For optimal effectiveness, preparations and supplements should be consumed regularly, with a recommended minimum frequency of 3–5 servings distributed throughout the day [2].

Physical exercise significantly influences protein requirements but also some micronutrients. Individuals with PKU, who typically restrict foods of animal origin, may face challenges in obtaining adequate amounts of vitamin B12 through their diet alone. Given its crucial role in energy metabolism and overall health, especially for individuals engaged in endurance activities, addressing vitamin B12 supplementation included in the protein substitutes, becomes essential. Consequently, any changes to activity levels should entail careful re-evaluation of the quantity and timing of PKU supplementation throughout the day.

Amino acid mixtures are the main protein source for PKU patients, but isolated intact proteins, such as casein glycomacropeptides, also have potential benefits. The protein synthetic response to the ingestion of an isolated intact dietary protein is sustained over a longer time than the response to free-form amino acids due to the slower absorption of the amino acid component in dietary protein. Combining free-form amino acids with an intact protein is appealing due to the beneficial effects of both. The use of a balanced formulation of amino acids combined with a high-quality intact protein should rapidly lead to a significant increase in leucine concentrations, which would activate protein synthesis at a molecular level while ensuring prolonged availability of all necessary precursors for protein synthesis.

As specific recommendations on protein requirements for physically active individuals with PKU are lacking, individual needs should be determined using general guidelines for healthy individuals and periodic blood Phe level assessments. Deviations from optimal blood Phe levels in patients with PKU, especially adolescents and adults, often result in poor dietary adherence. Although few rigorous studies have analyzed adherence in PKU, some observational reports have identified associated barriers and behaviors [54,55]. While direct assessment of blood Phe levels is valuable, consensus is lacking on the number of measurements that need to be within target ranges and on the frequency and timing of measurements.

Diverse strategies have been proposed to enhance dietary adherence among patients with PKU, but there is no universally effective approach. Individualization is thus crucial to foster long-term adherence in this setting. An approach combining prescribed physical exercise and dietary supplementation is increasingly recognized as an effective means of enhancing adherence among these patients.

Personalized sports nutrition plans, including specific recommendations for protein substitutes and hydration strategies, optimize performance while managing Phe intake effectively. Recommending low-protein sports nutrition products such as energy bars, drinks, and recovery shakes that are specifically formulated for individuals with PKU, provide convenient options for fueling workouts and support post-exercise recovery. Behavioral techniques such as goal-setting and self-monitoring support adherence to dietary regimens, fostering positive habits and coping mechanisms. Educational programs and workshops provide essential knowledge on PKU management in the context of physical activity, empowering individuals to make informed choices about nutrition and hydration. Utilizing technological aids, such as mobile applications and online platforms, facilitates tracking phenylalanine intake and accessing personalized nutritional information. By implementing these multifaceted approaches, healthcare providers can effectively support individuals with PKU in adhering to their dietary regimen during sports and physical activities, promoting optimal performance and overall well-being.

Achieving effective nutritional management of PKU in athletes and sports practitioners is challenging but possible. Existing recommendations must be interpreted with care, and further research is needed to better understand the impact of different protein substitutes on post-exercise protein synthesis. Further analysis should explore the effects of chronic exercise on blood Phe control and natural protein tolerance in PKU. Elite athletes with PKU obtain most of their protein (up to 100 g daily) from protein substitutes. These supplements should be strategically distributed throughout the day, in line with dietary principles for elite athletes, and with an emphasis on protein–carbohydrate substitution [53]. Post-exercise supplementation should be designed to improve recovery and muscle hypertrophy. The overarching goal is to achieve maximal athletic performance without compromising optimal blood Phe control.

## 7. Timing of Protein Intake for Exercise

Timing of protein intake in relation to exercise might be critical in some individuals. Intake can be organized around training or regular meals. Although the term “food intake” is used because it is the standard term to describe the consumption of nutrients, in PKU it refers to protein substitutes or permitted foods.

Recommendations on the distribution of protein intake in relation to exercise are divided into three categories: before, during, and after exercise. The goal of pre-exercise nutrition is to ensure sufficient glycogen stores to provide energy over prolonged periods of activity. Consuming slow-absorbing carbohydrates approximately 2 h before exercise stabilizes blood glucose levels and enhances performance. Lipids and proteins are not mandatory before exercise, but they do have an important role in post-exercise replenishment. Strategies such as “train low, compete high” are designed to optimize lipid availability during exercise, but the long-term effects of high-fat diets remain controversial. 

Adequate hydration, primarily from water, is important, and pre-hydration may be warranted in certain situations. Hydration is critical for individuals engaging in exercise to maintain performance and prevent dehydration. Pre-hydration involves consuming fluids 500–600 mL 2–3 h before exercise, followed by an additional intake of 250–300 mL 10–20 min before. During exercise, regular fluid intake of approximately 200–300 mL every 10–20 min replenishes lost fluids and maintains hydration. Post-exercise, individuals should continue hydrating, aiming for 500–700 mL for every 0.5 kg lost during exercise. Monitoring hydration status through thirst cues, urine color, and body weight changes helps ensure adequate fluid intake. Sports drinks with electrolytes and carbohydrates can be beneficial for longer or higher-intensity workouts. Personalized hydration strategies, considering factors like body size, sweat rate, and environmental conditions, are key to maintaining optimal hydration levels and supporting performance during exercise [56,57].

Maximization of glycogen stores through consumption of slow-absorbing carbohydrates is crucial in the days leading up to a competition. Maintenance of adequate hydration is equally important. Dietary adjustments are not necessary for regular pre-competition training. Athletes should have their last meal 2–3 h before competing or exercising, taking care to avoid new dishes or excessive condiments, and focusing on liquid foods in the last hour [58]. Pre-exercise nutrition is also influenced by the time of the event. Slow-absorbing carbohydrates consumed the night before, for example, may be beneficial for early morning exercise.

The need to replenish nutrients during exercise depends on the duration of the activity. Short, intense activities may not require replenishment, while activities exceeding 1 h may require the intake of sports drinks or solid elements to restore carbohydrates and minerals. Balancing liquid and carbohydrate intake is crucial to avoid compromising fluid absorption [59].

Post-exercise nutritional goals include replenishment of lost nutrients, with an emphasis on fluids, micronutrients, and, depending on the duration of the activity, carbohydrates. Energy drinks or fruit are recommended. Early lipid consumption after exercise is discouraged as it can interfere with carbohydrate absorption. Adequate protein intake after exercise, often facilitated by protein supplements, supports muscle development. Antioxidants are also important post-exercise nutrients as they counteract oxidative stress caused by free radicals [60].

Overall daily protein intake usually accounts for 10–15% of daily energy intake, but individual needs vary according to a range of factors, including type of activity, muscle mass, and glycogen stores. Athletes are generally advised to consume 1.2–1.8 g of protein/kg/d to maintain muscle mass, with adjustments for specific goals. A protein intake of around 1.5 g/kg, even in strength/strength sports, is considered adequate as reaching higher values becomes difficult. Small amounts of protein foods containing Phe can be consumed depending on the severity, but excessive protein intake (>2 g/kg/day), especially with depleted glycogen stores, may lead to increased ketone body and urea concentrations, potentially causing early dehydration when disbalanced products or inappropriate additives are used. Individualized protein calculations based on specific needs are crucial for optimizing athletic performance and managing PKU-related nutrition. Both in PKU and non-PKU individuals, the intake of ad hoc products is necessary. Key food intake recommendations for different phases of exercise are summarized in Table 2, with a focus on tailored approaches and considerations for individuals with PKU.

## 8. Adherence to Supplements and/or Protein Substitutes

Individuals with PKU may find it challenging to adhere to supplement regimens and protein substitutes while managing Phe levels to prevent associated complications. When considering the dietary management of PKU in athletes, finding a balance between optimizing athletic performance and maintaining appropriate blood Phe levels is crucial. We emphasize the importance of a personalized approach to dietary planning, which may involve experimenting with different strategies and adjusting intake levels based on individual responses and blood Phe monitoring. We highlight the need for flexibility and ongoing assessment, to empower athletes with PKU to optimize their performance while effectively managing their condition.

Adherence requires a multifaceted strategy rooted in education, personalization, monitoring, integration of supplements into daily routines, behavioral support, simplification, incentives, social support, involvement of healthcare staff, and customization. Education plays a central role as it helps patients understand the purpose of supplementation and its broader health benefits. Tailoring regimens to individual preferences, utilizing reminders, and integrating supplements into daily routines can all enhance adherence. Finally, adherence can also be improved by family, social, and healthcare support and involvement [61].

Collaborating with a registered dietitian specialized in metabolic disorders is important for developing personalized nutrition plans that incorporate the protein substitutes aligned with taste preferences and nutritional needs. Supplementation with low-protein foods, fortified substitutes, and creative recipe exploration is essential. Finally, it is crucial to address psychological and social factors through support groups and counseling, and to recognize the challenges of dietary restrictions.

In the same term, when evaluating the use of nutritional supplements in non-PKU elite athletes, it has been shown that the adherence to supplement regimens is influenced by multiple factors, from knowledge dissemination to regulatory considerations, palatability, peer behavior, and individual responses. In brief, adherence can be enhanced through comprehensive knowledge, accessibility, professional guidance, and in particular, tailored recommendations [62].

All patients with PKU, whether athletes or not, can benefit from consulting a healthcare specialist, such as a registered dietitian, a metabolic specialist, or a sports scientist. These professionals can help ensure that dietary adjustments are tailored to individual needs and health conditions to ensure effective PKU management and optimal physical performance. This comprehensive approach aims to improve adherence and ultimately overall health outcomes in both athletes and non-athletes with PKU.

## 9. Conclusions

The paper addresses the complex dietary requirements of individuals with PKU. PKU is managed with dietary protein restrictions and in many cases supplementation with Phe-free amino acids or blends of glycomacropetides. These preparations, designed for optimal macronutrient and micronutrient content, cover most of an individual’s protein needs but unlike typical dietary sources of protein do not include Phe. Individuals with PKU have higher protein requirements than the general population due to lower protein availability in supplements and potential mitigation of protein catabolism.

Optimal physical development is feasible in PKU, regardless of disease severity or extent of dietary restrictions. Exercise is essential for a healthy lifestyle, and individuals with PKU are encouraged to engage in regular physical activity in line with WHO guidelines.

The paper advocates for personalized guidance and reinforcement of adherence through comprehensive knowledge and professional support. Proper adherence to substitute regimens is crucial, and lessons can be learned from the field of sports medicine. Strategically-timed intake of protein substitutes within recommended safety limits can optimize muscle protein synthesis, enhance body composition, and improve sports performance. Individual factors must be taken into consideration to achieve optimal outcomes, and adherence can be reinforced through comprehensive knowledge, accessibility, and professional guidance to tailor recommendations to individual needs and conditions. Further research is required to explore how sports nutrition strategies applied to different sports and competition levels can enable athletes with PKU to achieve optimal performance.

Both athletes and non-athletes with PKU can benefit from advice from healthcare specialists, including registered dietitians, metabolic specialists, and sport scientists. This collaborative approach will help align dietary adjustments with individual health statuses, facilitating both effective PKU management and optimal physical performance. Comprehensive, tailored strategies are necessary to enhance adherence and overall health outcomes in individuals managing PKU or incorporating protein substitutes and supplements into athletic regimens.

## 10. Key Points

-Protein requirements in individuals with PKU are fulfilled with specially designed preparations, which, administered in 3–5 servings a day, provide up to 80% of daily requirements.-Exercise affects protein utilization and necessitates careful supplement timing. Amino acid mixtures are primary sources of protein, but intact proteins such as casein glycomacropeptides offer additional benefits.-Protein needs should be determined using general guidelines and periodic blood Phe level assessments. Deviations may indicate poor adherence.-A combined approach of prescribed exercise and dietary supplementation is ideal for enhancing adherence in patients with PKU.-Protein recommendations in relation to exercise vary according to the time of intake (before, during, or after exercise).-Pre-exercise recommendations include slow-absorbing carbohydrates for glycogen reserves.-Nutrient replenishment during exercise varies with the duration of the activity.-Post-exercise recommendations focus on nutrient replenishment, hydration, and adequate protein intake for recovery and muscle development.-Daily protein intake should account for 10–15% of total daily energy expenditure. A daily intake of approximately 1.5 g of protein/kg is recommended for athletes; excessive protein may lead to dehydration.

## Figures and Tables

**Table 1 nutrients-16-00639-t001:** Selected studies analyzing the impact of physical activity on body composition, metabolic control, and supplements (protein substitutes) in patients with PKU.

PKU-Related Variables Studied	Impact of Physical Activity	References	Brief Description of Study
Physical development	Optimal physical development regardless of PKU phenotype or severity of dietary restrictions. Proposed to avoid excessive weight gain.	Belanger-Quintana et al., 2011 [5]	Long-term anthropometric data collected from individuals with PKU
Body composition: body protein and bone density	Modified body composition and increased body protein and bone density	Allen et al., 1995 [11]	Assessment of resting energy expenditure showed similar levels in children with PKU and controls. Children with PKU had lower body protein and bone mineral density, suggesting potential predisposition to overweight due to altered body composition.
Body composition: weight, fat, and fat-free mass index	Lower weight and fat mass index and higher fat-free mass index	Jani et al., 2017 [12]	Analysis of protein intake (total protein, intact protein, and medical foods) and body composition in patients with PAH deficiency. Protein intake was assessed using food records. Body composition was measured through DXA, including fat-free mass index and fat mass index. Physical activity levels (light vs. intense) were evaluated using questionnaires.
Lower energy expenditure	Increased resting energy expenditure	Allen et al., 1995 [11]; Quirk et al., 2010 [13]	Assessment of agreement between measured and predicted resting energy expenditure in female adolescents with PKU. Findings showed that predictive equations consistently underestimated measured resting energy expenditure.
Elevated Phe levels and Phe:Tyr ratio	No significant effect demonstrated. Further investigation required.	Grünert et al., 2013 [14]; Mazzola et al., 2015 [9]	Comparison of various parameters, including basal metabolic rate, peak oxygen consumption, and blood biomarkers between PKU patients and matched controls. Phe and Tyr levels were also evaluated in adult patients with PKU following acute aerobic exercise.
Sports nutrition and protein substitute intake	Need to adapt protein substitute intake to different clinical scenarios	Rocha et al., 2019 [10]	Recommendation of tailored strategies to enhance performance, considering impact of endurance exercise on blood Phe levels. The authors proposed optimizing protein intake through Phe-free L-amino acid supplements for strength training, with an emphasis on post-exercise protein and carbohydrate intake. Additional benefits of Phe-free supplements during intermittent exercise were noted. Attention should be paid to weight management and carbohydrate intake in aerobic exercise and the incorporation of diluted Phe-free supplements during exercise.
Protein requirements to balance nitrogen loss and maintain body protein mass in individuals with moderate physical activity levels	Major impact on protein metabolism; additional protein may be necessary to support global energy demands.	van Wegberg et al., 2017 [2]	Authors were unable to recommend additional protein requirements for high-level sports due to a lack of evidence in the field of PKU
Patients with PKU had lower hand grip strength than matched controls	Increased muscle mass	Rojas-Agurto et al., 2023 [15]	Comparison of muscle mass, function, and bone health in young adults with PKU across three groups: those using a Phe-free protein substitute; those on a mostly vegan diet after using the substitute up to 18 years of age; healthy controls. Variables assessed included dietary recall, blood parameters, body composition, bone mineral density, rectus femoris thickness, grip strength, submaximal exercise test, and walking speed.
PKU guidelines	Patients with PKU should be encouraged to do at least 30 to 45 min of physical activity per day for general health. Optimize diet, regarding the intensity of physical activity.	MacDonald et al., 2020 [16]	General recommendations of daily exercise for patients with PKU: 30 to 45 min, for a total of at least 300 min per week. General diet recommendations for athletes with PKU: high carbohydrate diets, prioritizing carbohydrate-rich foods before and after exercise, ensuring proper hydration, and incorporating a protein substitute dose during the immediate post-exercise recovery phase.

Abbreviations: DXA, Dual-energy X-ray absorptiometry for Bone Density Scan; PAH, phenylalanine hydroxylase; Phe, phenylalanine; PKU, phenylketonuria; Tyr, tyrosine.

**Table 2 nutrients-16-00639-t002:** Timing of food intake in relation to exercise.

Segment	Recommendations
Before exercise	Prioritize slow-absorbing carbohydrates for sustained energy, avoid excesive loading lipids and proteins, and consider hydration strategies.
During exercise	Tailor nutrient replenishment to exercise duration, balancing liquid and carbohydrate intake to maintain fluid absorption.
After exercise	Emphasize post-exercise nutrient replenishment, including fluids, micronutrients, carbohydrates, and protein or protein substitutes. Discourage early lipid consumption and highlight the importance of antioxidants.
Overall daily protein intake	Aim for 10–15% of daily energy intake. Athletes are advised to consume 1.2–1.8 g of protein/kg/day. Caution against excessive protein intake or the use of innarpopiated and disbalanced products, especially with depleted glycogen stores, to prevent potential dehydration. Individualized protein calculations are crucial for optimizing athletic performance and managing PKU-related dietary considerations.

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
