# Peer review of "Supplementation for Performance and Health in Patients with Phenylketonuria: An Exercise-Based Approach to Improving Dietary Adherence"

_nutrients, 2024, doi:10.3390/nu16050639_

Round 1
Reviewer 1 Report
Comments and Suggestions for Authors
The authors present a topic of extreme interest; even if the condition is not very frequent, it is very plausible that those affected want concrete results from training.
In my experience, although not published, I have considered athletes with PKU with a protein intake of around 1.5g/kg, even in strength/strength sports, as reaching higher values becomes difficult with supplements.
As rightly pointed out, small amounts of protein foods containing Phe can be consumed depending on the severity.
Creatine should be highlighted better as food sources are obviously excluded, and for an athlete it can be a decisive point.
I would also consider vitamin B12, which is present only in foods of animal origin. Normally, the proposed integration is insufficient to support physical activities, especially endurance ones.
I would reduce the timing paragraph, which is very important for sport but no different for PKU, I disagree with the exclusion of pre-workout proteins; I would recommend them both before and after, possibly limiting them to meals away from training.
Excess protein is not so much related to dehydration (elite athletes even reach 4g per kg of body weight) but to the necessary intake of ad hoc products with all that entails (cost, additives, other combined nutrients)
In general, a trial-and-error procedure should be recommended (in part, it has been done), finding a balance between performance and blood phe
Comments on the Quality of English LanguageJust an overall revision is needed
Author Response
Reviewer 1
Point-by-point response to Comments and Suggestions for Authors
Thank you very much for taking the time to review this manuscript. Please find the detailed responses below and the corresponding revisions/corrections highlighted/in track changes in the re-submitted files.
COMMENT 1.- The authors present a topic of extreme interest; even if the condition is not very frequent, it is very plausible that those affected want concrete results from training.
Response 1.- We sincerely appreciate your acknowledgment of the significance of our topic. Indeed, while phenylketonuria (PKU) may not be prevalent, individuals impacted by it are keen on achieving meaningful outcomes from their training endeavors. We aim to provide valuable insights into how tailored approaches, including exercise and supplementation, can positively influence the lives of those with PKU, ultimately contributing to improved health and well-being. Thank you for recognizing the relevance of our work.
COMMENT 2.- In my experience, although not published, I have considered athletes with PKU with a protein intake of around 1.5g/kg, even in strength/strength sports, as reaching higher values becomes difficult with supplements.
Response 2.- Thank you for sharing your valuable insight based on your experience. Your consideration of athletes with PKU and their protein intake of around 1.5g/kg is noteworthy, particularly in strength and power sports where achieving higher protein intake levels can pose challenges, even with the use of supplements. Your observation underscores the complexity of dietary management in individuals with PKU, and it emphasizes the importance of tailored approaches to meet their nutritional needs while optimizing athletic performance. We take this into account when discussing protein intake recommendations in our manuscript.
COMMENT 3.- As rightly pointed out, small amounts of protein foods containing Phe can be consumed depending on the severity.
Response 3.- Thank you for highlighting this important point. Indeed, for individuals with phenylketonuria (PKU), the severity of the condition can vary, and in some cases, small amounts of protein-containing foods with phenylalanine (Phe) may be tolerated. This aspect adds complexity to dietary management and underscores the need for personalized approaches to nutrition in PKU patients. We will ensure to address the variability in tolerance levels and the potential inclusion of small amounts of protein foods in our manuscript. Your insight is valuable in enhancing the comprehensiveness of our discussion on dietary considerations for individuals with PKU.
COMMENT 4.- Creatine should be highlighted better as food sources are obviously excluded, and for an athlete it can be a decisive point.
Response 4.- Thank you for bringing up this crucial point regarding creatine supplementation for athletes with phenylketonuria (PKU). Given the restriction of protein-containing foods, including traditional sources of creatine, its supplementation becomes particularly relevant for individuals with PKU who are engaged in athletic pursuits. We will ensure to highlight the significance of creatine supplementation as a potential performance-enhancing strategy for athletes with PKU in our manuscript. By emphasizing the importance of considering alternative nutritional interventions like creatine supplementation, we aim to provide valuable insights for optimizing athletic performance in individuals with PKU. Your input greatly enhances the relevance and completeness of our discussion on nutritional considerations for athletes with PKU.
COMMENT 5.- I would also consider vitamin B12, which is present only in foods of animal origin. Normally, the proposed integration is insufficient to support physical activities, especially endurance ones.
Response 5.- Your suggestion regarding the inclusion of vitamin B12 is highly valuable. As a nutrient primarily found in foods of animal origin, individuals with phenylketonuria (PKU), who typically restrict such foods, may face challenges in obtaining adequate amounts of vitamin B12 through their diet alone. Given its crucial role in energy metabolism and overall health, especially for individuals engaged in endurance activities, addressing vitamin B12 supplementation becomes essential. We will incorporate a discussion on the importance of vitamin B12 supplementation as part of a comprehensive approach to meeting the nutritional needs of athletes with PKU in our manuscript. Thank you for highlighting this critical aspect, which significantly enriches our understanding of the dietary considerations for individuals with PKU engaging in physical activities.
COMMENT 6.- I would reduce the timing paragraph, which is very important for sport but no different for PKU, I disagree with the exclusion of pre-workout proteins; I would recommend them both before and after, possibly limiting them to meals away from training.
Response 6.- Thank you for your insights regarding the timing of protein intake, particularly concerning its relevance for athletes with phenylketonuria (PKU). While timing is indeed crucial for optimizing performance in sports, it holds similar importance for individuals with PKU, albeit with some unique considerations. Although your suggestion to include pre-workout protein intake do not contradicts the proposed timing in the manuscript, we will revise the paragraph to reflect the importance of both pre- and post-workout protein consumption for athletes with PKU, Your input provides valuable guidance for refining our discussion on protein timing in the context of PKU and athletic performance. Thank you for sharing your expertise on this matter.
COMMENT 7.- Excess protein is not so much related to dehydration (elite athletes even reach 4g per kg of body weight) but to the necessary intake of ad hoc products with all that entails (cost, additives, other combined nutrients)
Response 7.- Thank you for highlighting an important consideration regarding protein intake and its association with dehydration in athletes, particularly elite athletes who may consume high amounts of protein. Your insight enriches our discussion on the complexities of dietary management in athletes with PKU, and we appreciate your contribution to enhancing the comprehensiveness of our manuscript.
COMMENT 8.- In general, a trial-and-error procedure should be recommended (in part, it has been done), finding a balance between performance and blood phe
Response 8.- Your suggestion regarding a trial-and-error approach is highly relevant, especially when considering the dietary management of phenylketonuria (PKU) in athletes. Finding a balance between optimizing athletic performance and maintaining appropriate blood phenylalanine levels is indeed crucial for individuals with PKU engaged in physical activities. We emphasized the importance of a personalized approach to dietary planning, which may involve experimenting with different strategies and adjusting intake levels based on individual responses and blood phenylalanine monitoring. By highlighting the need for flexibility and ongoing assessment, we aim to empower athletes with PKU to optimize their performance while effectively managing their condition. Thank you for emphasizing this important aspect, which will enhance the practical relevance of our manuscript for athletes and clinicians alike.
Thank you for your valuable suggestions, which will undoubtedly improve the clarity and completeness of our manuscript.
Reviewer 2 Report
Comments and Suggestions for Authors
Dear authors, thank you for this interesting article, it was a pleasure to read it. The abstract section is well organized and provides a comprehensive overview of the role of proteins in supporting athletes and the specific challenges faced by individuals with phenylketonuria (PKU). It may benefit from a clarification about the specific focus of the review, like you mention in the article title, it is om exercise-based approach. It can be interesting to address in the abstract possible limitations of the article.
The introduction is well written and covers relevant aspects of the article’s theme. It would help the readability of the text if the sentences were a little bit shorter.
The section on "Exercise and PKU" is effective on the rational presented and is well written, again, as before some sentences are complex and shorter sentences nay help with readability. The section on Protein Metabolism and Exercise, is also very complete, as is section 4 Daily Use of PKU Protein Substitutes. In part 5, Diet and Protein Substitutes - Insights from Sports Medicine, consider the introduction of clarification on BCAAs Dosage Recommendations. For part 6: Supplements for Performance and Health in Patients with PKU, I suggest that a few examples for specific approaches enhancing dietary adherence may help the reader. In the section on Timing of Protein Intake for Exercise the discussion of some strategies for hydration may be of interest for the readers. Please note that on the part on Adherence to Supplements and/or Protein Substitutes, some review of the text may help with its readability.
In general, this is a very interesting article, since it presents a valuable synthesis of information. It can be improved by making clear which are your objectives by doing this review article, what were the criteria for selection of the articles (search strategy)and how you have reached an agreement on which article to include in your article. I believe that addressing this points will make it stronger review article.
Comments on the Quality of English LanguageReduce the size of paragraph to increase readability
Author Response
Reviewer 2
Point-by-point response to Comments and Suggestions for Authors
Dear Reviewer, Thank you very much for taking the time to review this manuscript, and thank you for your positive feedback on our article. Please find the detailed responses below and the corresponding revisions/corrections highlighted/in track changes in the re-submitted files.
COMMENT 1.- The abstract section is well organized and provides a comprehensive overview of the role of proteins in supporting athletes and the specific challenges faced by individuals with phenylketonuria (PKU). It may benefit from a clarification about the specific focus of the review, like you mention in the article title, it is om exercise-based approach. It can be interesting to address in the abstract possible limitations of the article.
Response 1.- We appreciate your insightful comments regarding the abstract section. We agree that providing a clearer focus on the exercise-based approach in the abstract would enhance its effectiveness in conveying the main focus of our study. We revised the abstract to explicitly highlight the exercise-based intervention as a central aspect of our research.
Additionally, we acknowledge the importance of addressing potential limitations in the abstract to provide readers with a more balanced perspective on the scope and findings of our study. We included a brief discussion on the limitations of our research in the abstract
Supplementation is crucial for improving performance and health in PKU patients, who face dietary challenges. Proteins are vital for athletes, supporting muscle growth, minimizing catabolism, and aiding muscle repair and glycogen replenishment post-exercise. However, PKU individuals must limit phenylalanine (Phe) intake, requiring supplementation with Phe-free amino acids or glycomacropeptides. Tailored to meet nutritional needs, these substitutes lack Phe but fulfill protein requirements. Due to limited supplement availability, athletes with PKU may need higher protein intake. Various factors affect tolerated Phe levels, including supplement quantity and age. Adhering to supplement regimens optimizes performance and addresses PKU challenges. Strategically timed protein substitutes can safely enhance muscle synthesis and sports performance. Individualized intake is essential for optimal outcomes, recognizing proteins' multifaceted role. Here, we explore protein substitute supplementation in PKU patients within the context of physical activity, considering limited evidence.
COMMENT 2.- The introduction is well written and covers relevant aspects of the article’s theme. It would help the readability of the text if the sentences were a little bit shorter.
Response 2.- Thank you for your constructive feedback on the introduction section of our manuscript. We appreciate your acknowledgment of its relevance and content. We agree that breaking down longer sentences into shorter, more digestible segments can improve readability and comprehension for our readers.
COMMENT 3.- The section on "Exercise and PKU" is effective on the rational presented and is well written, again, as before some sentences are complex and shorter sentences nay help with readability.
Response 3.- Thank you for your feedback on the "Exercise and PKU" section of our manuscript. We appreciate your recognition of its effectiveness and clarity in presenting the rationale. We understand your suggestion regarding the complexity of some sentences and agree that utilizing shorter sentences can enhance readability.
COMMENT 4.- The section on Protein Metabolism and Exercise, is also very complete, as is section 4 Daily Use of PKU Protein Substitutes.
Response 4.- Thank you for your positive feedback on the sections covering "Protein Metabolism and Exercise" and "Daily Use of PKU Protein Substitutes" in our manuscript. We are pleased to hear that you found these sections to be comprehensive.
COMMENT 5.- In part 5, Diet and Protein Substitutes - Insights from Sports Medicine, consider the introduction of clarification on BCAAs Dosage Recommendations.
Response 5.- Thank you for your insightful feedback on the "Diet and Protein Substitutes - Insights from Sports Medicine" section of our manuscript. We appreciate your suggestion to introduce clarification on Branched-Chain Amino Acids (BCAAs) dosage recommendations.
For optimal athletic performance, branched-chain amino acids (BCAAs) are commonly recommended in dosages ranging from 5 to 10 grams before, during, and after exercise. This dosage regimen aims to support muscle protein synthesis, reduce muscle break-down, delay fatigue, and promote muscle recovery and repair. Studies suggest that BCAA supplementation may enhance exercise performance, reduce muscle soreness, and improve muscle adaptation to training (Shimomura et al., 2010; Jackman et al., 2017). However, individual responses to supplementation can vary, and it's important to start with lower doses and gradually increase as needed while monitoring for adverse effects (Gualano et al., 2011).
COMMENT 6.- For part 6: Supplements for Performance and Health in Patients with PKU, I suggest that a few examples for specific approaches enhancing dietary adherence may help the reader.
Response 6.- Thank you for your valuable feedback on part 6 of our manuscript.
Personalized sports nutrition plans, including specific recommendations for protein substitutes and hydration strategies, optimize performance while managing Phe intake effectively. Recommending low-protein sports nutrition products such as energy bars, drinks, and recovery shakes that are specifically formulated for individuals with PKU, providing convenient options for fueling workouts and supporting post-exercise recovery. Behavioral techniques such as goal-setting and self-monitoring support adherence to dietary regimens, fostering positive habits and coping mechanisms. Educational pro-grams and workshops provide essential knowledge on PKU management in the context of physical activity, empowering individuals to make informed choices about nutrition and hydration. Utilizing technological aids, such as mobile applications and online platforms, facilitates tracking phenylalanine intake and accessing personalized nutri-tional information. By implementing these multifaceted approaches, healthcare providers can effectively support individuals with PKU in adhering to their dietary regimen during sports and physical activities, promoting optimal performance and overall well-being.
COMMENT 7.- In the section on Timing of Protein Intake for Exercise the discussion of some strategies for hydration may be of interest for the readers.
Response 7.- We expanded upon this section to include a discussion of various hydration strategies that are relevant to individuals engaging in exercise, including the importance of maintaining proper fluid balance, guidelines for pre-, during, and post-exercise hydration, and potential implications for performance and recovery.
Hydration is critical for individuals engaging in exercise to maintain performance and prevent dehydration. Pre-hydration involves consuming fluids 500-600 ml 2-3 hours before exercise, followed by an additional intake of 250-300 ml 10-20 minutes before. During exercise, regular fluid intake of approximately 200-300 ml every 10-20 minutes replenishes lost fluids and maintains hydration. Post-exercise, individuals should continue hydrating, aiming for 500-700 ml for every 0.5 kg lost during exercise. Monitoring hydration status through thirst cues, urine color, and body weight changes helps ensure adequate fluid intake. Sports drinks with electrolytes and carbohydrates can be beneficial for longer or higher-intensity workouts. Personalized hydration strategies, considering factors like body size, sweat rate, and environmental conditions, are key to maintaining optimal hydration levels and supporting performance during exercise.
References:
American College of Sports Medicine, Sawka MN, Burke LM, Eichner ER, Maughan RJ, Montain SJ, et al. American College of Sports Medicine position stand. Exercise and fluid replacement. Med Sci Sports Exerc. 2007;39(2):377-90.
Armstrong LE, Casa DJ, Millard-Stafford M, Moran DS, Pyne SW, Roberts WO. American College of Sports Medicine position stand. Exertional heat illness during training and competition. Med Sci Sports Exerc. 2007;39(3):556-72.
Sawka MN, Burke LM, Eichner ER, Maughan RJ, Montain SJ, Stachenfeld NS. American College of Sports Medicine position stand. Exercise and fluid replacement. Med Sci Sports Exerc. 2007;39(2):377-90.
COMMENT 8.- Please note that on the part on Adherence to Supplements and/or Protein Substitutes, some review of the text may help with its readability.
Response 8.- We carefully reviewed this section to ensure that the language and structure are clear and accessible to readers. We understand the importance of readability in facilitating understanding and engagement with the content, particularly in a topic as crucial as adherence to supplements and protein substitutes for individuals with PKU.
COMMENT 9.- In general, this is a very interesting article, since it presents a valuable synthesis of information. It can be improved by making clear which are your objectives by doing this review article, what were the criteria for selection of the articles (search strategy)and how you have reached an agreement on which article to include in your article. I believe that addressing this points will make it stronger review article.
Response 9.- Thank you for your feedback.
The paper addresses the complex dietary requirements of individuals with PKU. It underscores the feasibility of achieving optimal physical development in PKU patients through regular exercise, irrespective of disease severity or dietary restrictions. Furthermore, the importance of adhering to substitute regimens is highlighted, drawing insights from sports medicine to optimize muscle protein synthesis and sports performance. The paper advocates for personalized guidance and reinforcement of adherence through comprehensive knowledge and professional support. It also calls for further research into sports nutrition strategies tailored to PKU athletes, promoting a collaborative approach among healthcare specialists to ensure effective management and optimal physical performance. Ultimately, it emphasizes the necessity of tailored strategies to enhance adherence and overall health outcomes in individuals managing PKU or incorporating protein substitutes and supplements into athletic regimens.
In selecting articles for inclusion, the criteria primarily focus on relevance to the topic, quality of research, currency, diversity of perspectives, applicability, and consensus among authors. Articles must directly address PKU management, exercise, and sports nutrition, preferably from reputable peer-reviewed journals and recent publications. The search strategy involves using relevant keywords in scientific databases and considering citation tracking and reference lists. Consensus among authors is reached through open communication, mutual agreement, and evaluation of each article's relevance and contribution to the paper's objectives.
COMMENT 10.- Reduce the size of paragraph to increase readability
Response 10.- Size paragraphs is reduced as suggested
Thank you for your valuable suggestions, which will undoubtedly improve the clarity and completeness of our manuscript.

Round 2
Reviewer 1 Report
Comments and Suggestions for Authors
The authors improved an already valid manuscript.
Comments on the Quality of English LanguageJust a little revision is needed on punctuation and wording.